# The Relevance of Serum Macrophage Migration Inhibitory Factor Level and Executive Function in Patients with White Matter Hyperintensity in Cerebral Small Vessel Disease

**DOI:** 10.3390/brainsci13040616

**Published:** 2023-04-05

**Authors:** Jianhua Zhao, Xiaoting Wang, Miao Yu, Shiyun Zhang, Qiong Li, Hao Liu, Jian Zhang, Ruiyan Cai, Chengbiao Lu, Shaomin Li

**Affiliations:** 1Henan Joint International Research Laboratory of Neurorestoratology for Senile Dementia, Henan Key Laboratory of Neurorestoratology, Neurology Department, First Affiliated Hospital of Xinxiang Medical University, Xinxiang 453100, China; 2Imaging Department, First Affiliated Hospital of Xinxiang Medical University, Xinxiang 453100, China; 3Sino-UK Joint Laboratory of Brain Function and Injury of Henan Province, Department of Physiology and Neurobiology, Xinxiang Medical University, Xinxiang 453003, China; 4Ann Romney Center for Neurologic Diseases, Brigham and Women’s Hospital, Harvard Medical School, Boston, MA 02115, USA

**Keywords:** cerebral small vessel disease, executive function, macrophage migration inhibitory factor, white matter hyperintensity

## Abstract

(1) Objective: To investigate the relationship between serum macrophage migration inhibitory factor (MIF) level and white matter hyperintensity (WMH) and executive function (EF) in cerebral small vascular disease (CSVD), and assess the impact and predictive value of MIF level and Fazekas scores in CSVD-related cognitive impairment (CI) (CSVD-CI); (2) Methods: A total of 117 patients with WMH admitted to the First Affiliated Hospital of Xinxiang Medical College from January 2022 to August 2022 were enrolled. According to the Montreal cognitive assessment (MoCA) scale, subjects were divided into a normal cognitive group and an impaired group. All subjects required serum MIF level, 3.0 T MRI, and neuropsychological evaluation to investigate the risk factors for CDVD-CI, analyze the correlation between MIF level, WMH, and EF, and to analyze the diagnostic value of MIF and WMH degree in predicting CSVD-CI; (3) Results: 1. Fazekas score and MIF level were the risk factors of CSVD-CI. 2. The Fazekas score was negatively correlated with MoCA score, positively correlated with Stroop C-Time, Stroop C-Mistake, Stroop interference effects (SIE)-Time, SIE-Mistake, and color trails test (CTT) interference effects (CIE) (B-A). 3. The MIF level was positively correlated with Fazekas score, Stroop C-Time, SIE-Time, CTT B-Time, and CIE (B-A), and negatively correlated with MoCA score. 4. Fazekas score and MIF level were significant factors for diagnosing CSVD-CI; (4) Conclusion: The Fazekas score and MIF level may be the risk factors of CSVD-CI, and they are closely correlated to CI, especially the EF, and they have diagnostic value for CSVD-CI.

## 1. Introduction

Cerebral small vascular disease (CSVD) is one of the common, chronic, and progressive cerebrovascular diseases, accounting for about 25% of ischemic strokes, and it is also an important pathogen of dementia [1]. CSVD is caused by various pathological changes of intracranial arterioles, venules, and capillaries, with clinical manifestations of ischemic stroke, dementia, gait disturbance, urinary incontinence, depression, etc.

CSVD is one of the common causes of vascular cognitive impairment (VCI) and is associated with progressive cognitive decline and the emergence of new cognitive impairment [2]. VCI caused by CSVD accounts for 15–30% of clinical dementia cases, second only to Alzheimer’s disease (AD) [3,4]. The early clinical manifestations of CSVD-related cognitive impairment (CI) (CSVD-CI) are insidious and atypical [3,5], often detected as executive dysfunction (ED), such as attention and inhibition function, cognitive flexibility, information processing speed, and visuospatial dysfunction [6,7,8], which easily affect activities of daily living, social participation, work performance, and functional prognosis [9], and ED is the core and first symptom of CI [10]. The assessment of executive function (EF) is an important part of the neuropsychological assessment of the elderly, has a high sensitivity to CI, and is a vital factor affecting the functional rehabilitation in stroke [11]. With the increase in population aging worldwide, CSVD and CI impose a significant burden on individuals and the society; therefore, early identification of EF impairment in CSVD is critical. Since the pathogenesis of CSVD and VCI remains unclear [1,12,13], the search for sensitive and accurate biomarkers will provide new scientific ideas.

Migration inhibitory factor (MIF) is an evolutionarily highly conserved low molecular homotrimeric protein (about 12.5 kDa). MIF is involved in various biological functions, including leukocyte recruitment, inflammation, immune response, cell proliferation, tumorigenesis, and regulation of glucocorticoids. In order to identify detection and treatment targets, studies have detected the lineage changes of various cytokines/chemokines in the plasma of ischemic stroke patients and confirmed that MIF is significantly elevated [14]. MIF could accelerate atherosclerosis (AS) through immune reaction, inflammation, and oxidative stress, and promote neuronal death, and thus aggravate the development of stroke [15]. MIF is associated with biomarkers of AD pathology and predicts cognitive decline in MCI and mild dementia [16]. CSVD is a small ischemic or bleeding lesion caused by pathological small vessels or brain degeneration [17], and is an important cause of ischemic stroke. CSVD has similar etiology and pathological mechanisms to stroke as hypoxia, inflammation, and immunoreaction, and has a complex relationship with AD [18,19]. It is speculated that MIF may act on CSVD and CI through different pathogeneses and become a reliable target for the detection and treatment of CSVD. Based on these, this article mainly studies the risk factors of CSVD-CI and the relevance between MIF and white matter hyperintensity (WMH) and executive dysfunction.

## 2. Materials and Methods

### 2.1. Subjects

A total of 117 patients with WMH in CSVD who were admitted to the Department of Neurology of the First Affiliated Hospital of Xinxiang Medical College from January 2022 to August 2022 were enrolled. Inclusion criteria: patients who (1) met the diagnostic criteria for CSVD with WMH; (2) had complete MRI imaging and clinical data; (3) had no communication or comprehension disorders; (4) had not taken nootropic drugs or immunosuppressants. Exclusion criteria: patients who (1) had suffered from nonvascular leukoencephalopathy (immune demyelinating, metabolism, toxicity, infectious); (2) had suffered from intracranial or extracranial large-vessel stenosis (stenosis degree >75%), or intracranial hemorrhage disease; (3) had history of ischemic stroke with infarction diameter > 15 mm or cardiogenic cerebral infarction; (4) had other central nervous system (CNS) diseases, such as CNS infection, epilepsy, multiple sclerosis, Parkinson’s syndrome, other inflammatory demyelinating diseases, brain trauma, and brain tumors; (5) had history of malignant tumors and serious cardiac and pulmonary diseases; (6) could not cooperate due to mental disease and severe language disorders; (7) had abnormal hearing or vision, color blindness, and color weakness.

Group criteria: According to the “2019 Chinese Guidelines for the Diagnosis and Treatment of Blood Vessel Cognitive Impairment” and the “Chinese Guidelines for the Diagnosis and Treatment of Cognitive Dysfunction Related to Cerebral Small Vessel Disease (2019)”, using the Montreal cognitive assessment—Beijing version (MoCA) [20]—subjects with scores below the adjusted criteria for education were defined as the WMH with impaired cognitive group (IC) (*n* = 65), and the remainder as the WMH with normal cognitive group (NC) (*n* = 52). The specific corrections are as follows: 19 points for 1–6 years of education, ≤24 points for 7–12 years, and <26 points for >12 years.

### 2.2. Methods

#### 2.2.1. Observational Indexes

Demographic data (gender, age, and educational qualifications). History of hypertension, diabetes, coronary heart disease, stroke, smoking (an average of 10 or more cigarettes per day for 5 years or more), and drinking (ethanol intake of 30 g or more per day for 5 years or more). Serum samples (total cholesterol (TC), triglyceride (TG), high-density lipoprotein (HDL), low-density lipoprotein (LDL), fasting blood glucose (FBG), serum creatinine (Scr), uric acid (UA), and homocysteine (Hcy)).

#### 2.2.2. Neuropsychological Test

The subjects were all tested by a trained physician using unified guidance in a quiet, ventilated, comfortable, and well-lit room, there are generally only 2 people with a tester and a subject to avoid interference from others. Overall cognitive function was assessed using the MoCA scale [20]. Executive function, attention, and information processing speed were assessed by Stroop test and color trails test (CTT). Language naming capabilities were evaluated by Boston naming test (BNT). Instrumental activities of daily living were assessed by instrumental activities of daily living scale (IADL).

##### Stroop Test

The Polish version of Victoria Stroop test (VST) [21] contains 3 cards: dot condition—Card D; word condition—Card W; interference condition (word printed in a different color)—Card C. Record indexes: For each condition, the completion time (Stroop-T) and the mistake number (Stroop-M) were compiled, and Stroop interference effects (SIE) were derived by calculating the time: Card C minus Card D (SIE-T), and the mistake number: Card C minus Card D (SIE-M). The SIE reflects the dominant inhibitory factor, the larger the SIE, the worse the interference suppression function, and the worse the EF.

##### CTT

The Polish version of color trails test (CTT) [22,23] is divided into two parts: CTT-A and CTT-B. Record indexes: For each part, the completion time (CTT-T) and the mistake number (CTT-M) are compiled. The part of A contains mistake number (CTT A-M), and the part of B contains mistake number (CTT B-M-N) and number of mistake colors (CTT B-M-C). CTT interference effects (CIE) are derived by calculating the time: Part B minus Part A (CIE (B-A)); the CIE reflects the fixed transfer factor, the greater the CIE, the more obvious the suppression of fixed shift.

#### 2.2.3. Evaluation of Cranial MRI

A 3.0 T magnetic resonance imaging (MRI) scanner from GE, USA, with an 8-channel head coil was used to perform conventional axial T1WI, T2WI, T2FLAIR, and DWI imaging on all subjects within 3 days of admission. The scan parameters in T1WI: slice thickness = 5.0 mm, slice spacing = 1.5 mm, repetition time (TR) = 1750 ms, echo time (TE) = Min Full, and matrix size = 320 × 192. T2WI: 5.0 mm/1.5 mm/3805 ms/110 ms/352 × 352. T2FLAIR: 5.0 mm/1.5 mm/8400 ms/120 ms/320 × 192. DWI: 5.0 mm/1.5 mm/3543 ms/minimum/162 × 192. The subjects’ imaging data were evaluated by 2 raters who were trained in MRI evaluation and ignored the clinical data. The evaluation of the WMH grade was performed with Fazekas classification method in magnetic resonance T2 FLAIR (T2 FLAIR) [24,25], in which WMH were divided into periventricular WMH (PVWMH) and deep WMH (DWMH) as follows: WMH typically occurs in white matter around the lateral ventricles (i.e., PVWMH), deep into the deep white matter and gray matter nuclei (i.e., DWMH), or as a bordered punctate sphere in deep cortical tissue. PVWMH scoring criteria: no WMH: 0 point; caps or pencil-thin lining: 1 point; smooth halo: 2 points; irregular PVWMH extending into the deep white matter: 3 points. DWMH scoring criteria: no WMH: 0 point; punctate foci: 1 point; beginning confluence of foci: 2 points; large confluent areas: 3 points. Total Fazekas score: PVWMH score + DWMH score. The main classification standard: grade mild WMH (1~2 points), grade moderate (3~4 points), grade severe (5~6 points). Presented in Figure 1.

#### 2.2.4. Determination of Serum MIF Level

An amount of 5 mL of elbow venous blood was sampled from all subjects in the morning within 24 h of admission in non-anticoagulant blood vessels, centrifuged at 3000 r/min for 10 min at room temperature, isolated the upper serum, and stored in a cryopreservation tube in a −80 °C freezer for later use. Enzyme-linked immunosorbent assay (ELISA) was used to detect serum MIF level (Wuhan Finn Biotechnology Co., Ltd., Wuhan, China) following the kit instructions.

#### 2.2.5. Statistical Analysis

The data were analyzed by SPSS 26.0 software (IBM, Armonk, NY, USA). In Table 1 and Table 2 are the data for continuous variables: the normal data were expressed as mean ± standard deviation (x ± s), and the non-normal data were expressed as the median and quartile; comparison between two groups was performed using independent *t*-test or Mann–Whitney U test. Data for categorical variables were expressed as percentages, and were compared by the Chi-square test or Fisher exact test. The risk factors for the occurrence of cognitive dysfunction in CSVD were analyzed by logistic univariate regression with statistically different and clinically significant variables as input variables in Table 3. Correlation analysis: In Table 4 and Table 5, Pearson correlation analysis was used for normal data and Spearman correlation was used for non-normal or categorical data. The receiver operating characteristic (ROC) curve analysis in Table 6 and Figure 2 was used to analyze the diagnostic value of serum MIF level and WMH degree in predicting CSVD-CI. *p* < 0.05 was considered statistically significant.

## 3. Results

### 3.1. Comparison of General Data in the Two Groups

The prevalence of hypertension in the IC group was higher than the NC group, and the difference was statistically significant (78.46% vs. 59.62%, *p* = 0.041). Results are presented in Table 1.

### 3.2. Comparison of Serum MIF Level, Total Fazekas Scores, and Cognitive Function Assessment in the Two Groups

In the IC group, the serum MIF level, total Fazekas score, paraventricular and deep WMH score, all evaluation indicators in the Stroop test, and CTT were higher than those in the NC group (*p* < 0.05), the BNT and IADL score were lower than the NC group (*p* < 0.05), and there exist statistically significant differences between the two groups, actual *p* values are presented in Table 2.

### 3.3. Logistic Regression Analysis of WMH-CI in CSVD

Logistic regression analysis was performed by taking hypertension, total Fazekas score, and MIF level as independent variables, and whether cognitive impairment was combined as a dependent variable, and the results showed that the total Fazekas score (OR = 1.422, 95% CI = 1.023~1.976, *p* = 0.036) and serum MIF level (OR = 1.007, 95% CI = 1.000~1.014, *p* = 0.038) were risk factors for whether CSVD patients were combined with CI. An increase of one point in the total Fazekas score suggested a 42.2% increased risk of CSVD-CI; an increase of 1 pg/mL MIF suggested a 0.7% increased risk of CSVD-CI. Results are presented in Table 3.

### 3.4. Correlation Analysis of WMH Degree and Cognitive Function

Correlation analysis showed that the total Fazekas score was negatively correlated with the total MoCA score and IADL, and it was positively correlated with Stroop D-T, Stroop W-T, Stroop C-T, Stroop W-M, Stroop C-M, SIE-T, SIE-M, and CTT A-T, CTT B-T, SIE (B-A), and actual *p* values are presented in Table 4. With the increase in the degree of WMH, the overall cognitive function and IADL decreased, most indicators in the Stroop test and CTT increased, and the executive function declined.

### 3.5. Correlation Analysis of MIF Level and WMH Degree and Cognitive Function

Correlation analysis showed that MIF level was positively correlated with the total Fazekas score, Stroop C-T, SIE-T, CTT B-T, and SIE (B-A), was negatively correlated with total MoCA score and BNT, and actual *p* values are presented in Table 5. With the increase in MIF level, the degree of WMH increased, some indicators in the Stroop test and CTT increased, executive function declined, and the overall cognitive function and IADL declined. 

### 3.6. The ROC Curve Analysis of the Diagnostic Value of Serum MIF Level and WMH Degree in Predicting CI in Patients with CSVD

ROC curve analysis showed that the area under the curve (AUC) of serum MIF level in the IC group was 0.661 (0.561~0.761), and the sensitivity and specificity were 0.723 and 0.596, respectively. The AUC of the total Fazekas score in the IC group was 0.658 (0.559~0.756), and the sensitivity and specificity were 0.462 and 0.846, respectively. MIF level *(p* = 0.003) and total Fazekas score *(p* = 0.003) were the significant factors for diagnosing CSVD-CI. Results are presented in Table 6.

## 4. Discussion

This is the first cohort study conducted in China to investigate the relationship between serum MIF level and WMH and EF. This study found that the total Fazekas score and serum MIF level were risk factors for CSVD-CI: compared to the NC group, an increase of one point in the Fazekas score suggested a 42.2% increased risk of CSVD-CI and an increase of 1 pg/mL MIF suggested a 0.7% increased risk of CSVD-CI in the IC group. As the Fazekas score increases, the more cognitive function is impaired, and with the increase in MIF level, the more severe the degree of WMH, the more severe the impairment to cognitive and executive function. Total Fazekas score and serum MIF level were found to be significant factors for diagnosing cognitive impairment due to CSVD.

In the present study, the results showed that the total Fazekas score was a risk factor for CI in patients with WMH, the degree of WMH was correlated with CI, and the ROC curve showed certain diagnostic value for CI in CSVD. Previous studies have shown that WMH is closely associated with the incidence of CI and dementia [26,27], mainly impairs EF, attention, information processing speed, memory and language function [28,29], and especially causes EF impairment [30]. An increase in total WMH load could predict cognitive decline, MCI, dementia, stroke, and even death [31], and the greater the WMH load, the more severe the cognitive impairment [32,33], which are consistent with the conclusion that the total Fazekas scores are negatively correlated with MoCA scores in this study. We found that in the Stroop test, with the increase in WMH load, except for the mistake number of Card D, the other indexes increased with varying degrees, especially SIE-T and SIE-M, representing the interference inhibition. Combined with the CTT test results, with the increase in WMH load, the time consumption and interference of Card A and Card B also increased, indicating that the patient’s information processing speed and fixed transfer ability were impaired, while the mistake numbers in Card A and Card B were not significantly correlated; this may because patients with CI try to extend reading time in exchange for reading correctness. These above results suggest that attention and ED may be more sensitive evaluation indexes in the early stage of CI. IADL involves complex activities related to the ability to live independently and can be monitored dynamically to assess CI [34], and the more severe the CI, the more obvious the degree of IADL disability. This study found that WMH severity was negatively correlated with IADL, suggesting that WMH lesions can cause decreased independent living ability. Early cognitive impairment of CSVD can be preventable and controllable. MRI should be performed as early as possible to assess the severity of WMH based on Fazekas, and to infer whether the patient has cognitive impairment and the severity of it. Timely screening of cognitive function with WMH, especially assessment of EF, and early intervention are expected to play a key role in the prognosis of CSVD patients.

This study found that MIF level was a risk factor for CI in patients with CSVD, and its level was positively correlated with cognitive function and the severity of WMH. Previous studies have found that MIF could affect the development of CSVD by affecting a variety of pathophysiological processes [35]. Arteriosclerosis is the main cause of chronic hypoxia hypoperfusion, while MIF is a key mediator of arteriosclerosis [36], participating in the entire process of arteriosclerosis by promoting leukocyte recruitment and damaging inflammation [37,38]. Previous studies have found that MIF is involved in the preclinical atherosclerosis process based on low-grade inflammation [39], and is associated with hypoendothelial function and increased vascular stiffness [40], while arteriosclerosis may be a common pathogenesis of CSVD [41]. Moreover, MIF can also promote the production of pro-inflammatory factors (IL-1β, IL-6, and intercellular cell adhesion molecule (ICAM)) [42,43]; aggravate many pathophysiological processes, including endothelial injury, blood–brain barrier(BBB) destruction, white matter lesions, etc. [44]; and participate in the whole process of inflammatory response. At the same time, MIF can promote endothelial autophagy induction, leading to endothelial barrier dysfunction [45], thereby increasing vascular permeability, and ultimately leading to the destruction of the BBB [14], and the degree of increased BBB permeability is associated with higher WMH load and cognitive decline [46]. Therefore, MIF can promote the occurrence and progression of CSVD through different pathogeneses, which in turn causes cognitive impairment.

This study showed that with the increase in MIF level, the total MoCA score decreased significantly (*p* = 0.001), and thus the increased MIF level may cause cognitive function decline. MIF level is positively correlated with the time consumption of Card C, SIE-T, SIE (B-A). Card C is the most difficult in the Stroop test, which can best reflect the ability to inhibit cognitive interference. SIE-T represents the dominant inhibitor and is the core factor of EF. SIE (B-A) represents the stereotype transfer factor, which can assess well the executive ability of the patients, and its sensitivity and specificity for the diagnosis of MCI caused by CSVD were 88% and 76%, respectively [47]. These above results show that the cognitive dysfunction influenced by MIF is mainly characterized by ED with impaired information processing speed, dominant inhibition ability, and static transfer ability, which is consistent with the characteristics of CSVD-CI, so CI caused by MIF may further affect cognitive function through cerebrovascular pathology. 

Although the ROC curve analysis in the present study showed that serum MIF level has a diagnostic value for CI in CSVD, the diagnostic sensitivity and specificity are still unsatisfactory and remained at 0.723 and 0.596, respectively. Because the sequence of ischemia, hypoxia, inflammatory response, and elevated MIF level is not well defined, the sensitivity of MIF is difficult to detect. MIF can be implicated in multiple CNS diseases through the inflammatory immune responses, such as stroke, neurodegeneration, multiple sclerosis, etc. Therefore, the specificity is difficult to detect. This may be related to the fact that MIF with other neurological factors, as diagnostic markers for CNS diseases, can further improve diagnostic specificity.

## 5. Limitations

There are some limitations to this study. First, the study was a cross-sectional study that only illustrated the health status of the patients at enrollment, and the older patients enrolled may have had a recall bias because of an inaccurate memory of their medical history. Second, the study included a small sample size and could not be carried out in a larger population. In the future, multicenter prospective cohort studies need to be carried out to explore in depth the role of MIF on the occurrence and development of CSVD and the mechanism of cognitive impairment related to CSVD.

## 6. Conclusions

The total Fazekas score and the increased serum MIF level may be the risk factors of CI in patients with CSVD, and they are closely related to CI, showing diagnostic value for CSVD-CI in the study. As Fazekas total score and MIF level increased, cognitive function was impaired and executive function declined significantly. Serum MIF level may be involved in the occurrence and development of CI in patients with CSVD, and executive function shows high sensitivity to CI; therefore, the detection of serum MIF level in CSVD can be conducive to the early identification of CI, so as to intervene early and improve prognosis.

## Figures and Tables

**Figure 1 brainsci-13-00616-f001:**
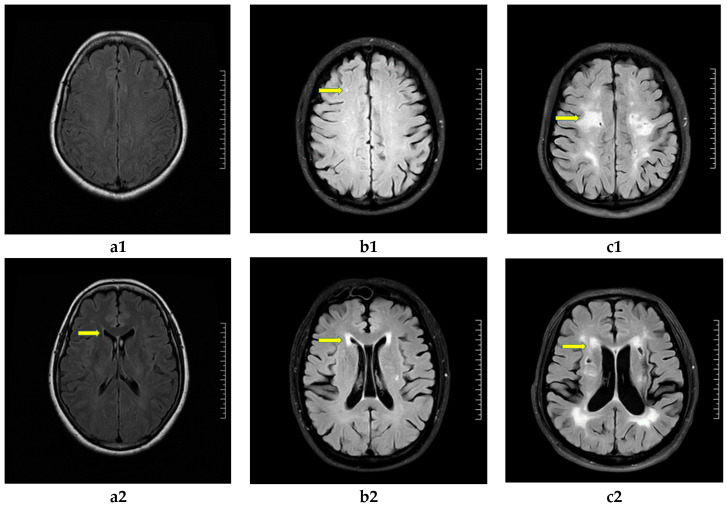
Neuroimaging characteristics and grading of white matter hyperintensity (WMH) in T2 FLAIR. Fazekas score: deep WMH (DWMH): (**a1**) (0 point), (**b1**) (1 point), (**c1**) (3 point). Periventricular WMH (PVWMH): (**a2**) (1 point), (**b2**) (2 point), (**c2**) (3 point). WMH grade: (**a1**,**a2**): grade mild; (**b1**,**b2**): grade moderate; (**c1**,**c2**): grade severe.

**Figure 2 brainsci-13-00616-f002:**
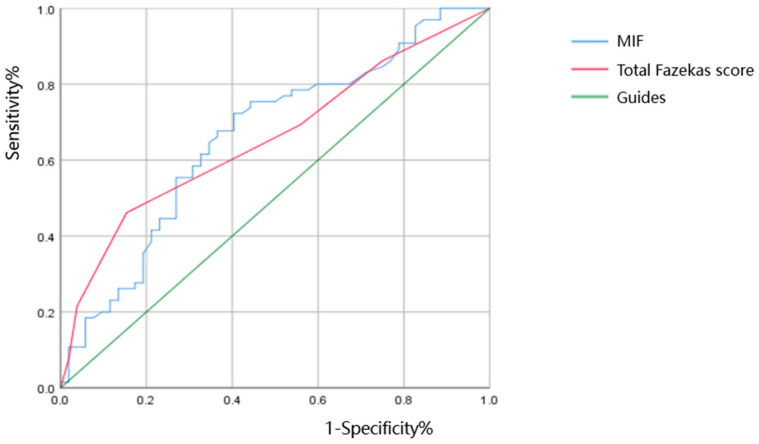
ROC curve of serum MIF level and WMH degree in predicting CSVD-CI.

**Table 1 brainsci-13-00616-t001:** Baseline Characteristics.

	NC Group (*n* = 52)	IC Group (*n* = 65)	χ^2^/t/u	*p*
Male proportion (%)	55.77%	52.31%	0.139	0.852
Age (years)	60.3 ± 8.8	61.6 ± 9.0	−0.820	0.414
BMI (Kg/m^2^)	24.81 (23.21, 27.47)	24.90 (22.82, 26.15)	−1.039	0.300
Education (years)	6.50 (5, 9)	8 (3.5, 9)	−0.480	0.634
Hypertension, *n* (%)	59.62%	78.46%	4.894	0.041
Diabetes, *n* (%)	25.00%	24.62%	0.002	1.000
CHD, *n* (%)	9.62%	7.69%	0.137	0.749
Stroke, *n* (%)	32.69%	29.23%	0.049	0.844
Smoke, *n* (%)	32.69%	32.70%	0.163	0.693
Alcohol, *n* (%)	26.92%	13.85%	3.128	0.102
TC (mmol/L)	4.25 (3.52, 5.26)	4.31 (3.40, 5.16)	−0.236	0.815
TG (mmol/L)	1.26 (0.91, 2.11)	1.25 (0.86, 1.74)	−0.771	0.443
HDL (mmol/L)	1.18 (1.06, 1.41)	1.13 (1.02, 1.28)	−1.405	0.161
LDL (mmol/L)	2.44 (1.79, 3.05)	2.38 (1.83, 3.06)	−0.167	0.869
FBG (mmol/L)	5.19 (4.76, 5.87)	4.88 (4.47, 5.84)	−1.171	0.243
Scr (µmol/L)	60.45 (52.43, 67.68)	64.10 (51.10, 70.40)	−0.570	0.571
UA (µmol/L)	261.50 (214.75, 305.00)	266.00 (216.50, 322.50)	−0.225	0.824
Hcy (µmol/L)	13.96 (11.07, 20.73)	14.27 (11.35, 19.64)	−0.154	0.879

Abbreviations: CHD: coronary heart disease; TC: total cholesterol; TG: triglycerides; HDL: high-density lipoprotein; LDL: low-density lipoprotein; FBG: fasting blood glucose; Scr: serum creatinine; UA: uric acid; Hcy: homocysteine.

**Table 2 brainsci-13-00616-t002:** Comparison of serum MIF level, total Fazekas score, and cognitive function assessment in the two groups.

	NC Group (*n* = 52)	IC Group (*n* = 65)	χ^2^/t/u	*p*
MIF (pg/mL)	139.50 (114.25, 180.75)	177.00 (141.50, 218.50)	−2.992	0.003
Total Fazekas score	3.00 (1.25, 3.00)	3.10 (2.00, 4.00)	−2.998	0.003
periventricular WMH	2.00 (1.00, 2.00)	2.10 (1.00, 2.00)	−2.476	0.013
deep WMH	1.00 (0.25, 1.00)	1.10 (1.00, 2.00)	−3.045	0.002
Stroop D-Time	21.00 (18.25, 28.00)	27.00 (20.00, 33.50)	−2.444	0.014
Stroop W-Time	30.50 (21.25, 38.75)	34.00 (29.00, 41.00)	−2.517	0.012
Stroop C-Time	38.00 (34.00, 52.50)	52.00 (39.00, 77.50)	−3.141	0.002
Stroop D-Mistake	0.00 (0.00, 0.00)	0.10 (0.00, 0.00)	−2.397	0.015
Stroop W-Mistake	0.00 (0.00, 0.00)	0.10 (0.00, 1.00)	−3.622	<0.001
Stroop C-Mistake	1.50 (0.00, 2.00)	3.00 (1.00, 4.00)	−2.605	0.009
SIE-Time	16.00 (10.00, 26.25)	26.00 (15.50, 43.00)	−2.527	0.011
SIE-Mistake	1.00 (0.00, 2.00)	2.00 (0.50, 4.00)	−2.178	0.029
CTT A-Time	74.00 (53.00, 108.75)	105.00 (67.50, 168.50)	−2.957	0.003
CTT B-Time	155.50 (120.50, 269.25)	266.00 (152.00, 428.00)	−3.475	<0.001
CIE (B-A)	97.00 (49.25, 141.50)	139.00 (90.00, 253.00)	−3.190	0.001
CTT A-Mistake	0.00 (0.00, 0.00)	0.10 (0.00, 1.00)	−2.958	0.003
CTT B-Mistake-Number	0.00 (0.00, 0.00)	0.10 (0.00, 1.00)	−2.184	0.028
CTT B-Mistake-Color	0.00 (0.00, 1.00)	0.10 (0.00, 2.00)	−2.363	0.018
BNT	21.96 ± 3.71	18.88 ± 3.63	4.529	<0.001
IADL	8.00 (8.00, 8.00)	7.90 (7.00, 8.00)	−3.621	<0.001

Abbreviations: SIE-Time: Stroop interference effects—Time; SIE-Mistake: Stroop interference effects—Mistake; CIE (B-A): color trails test interference effects—(TimeB-TimeA); BNT: Boston naming test; IADL: instrumental activities of daily living scale.

**Table 3 brainsci-13-00616-t003:** Regression analysis of WMH-CI risk factors.

	β	SE	Wald χ^2^	OR	95% CI	*p*
Hypertension	0.232	0.485	0.229	1.262	0.487~3.266	0.632
Total Fazekas score	0.352	0.168	4.401	1.422	1.023~1.976	0.036
MIF (pg/mL)	0.070	0.003	4.296	1.007	1.000~1.014	0.038

**Table 4 brainsci-13-00616-t004:** Correlation between WMH degree and cognitive function.

	r	*p*
Total MoCA score	−0.252	0.006
Stroop D-Time	0.234	0.011
Stroop W-Time	0.264	0.004
Stroop C-Time	0.334	<0.001
Stroop D-Mistake	0.070	0.454
Stroop W-Mistake	0.244	0.008
Stroop C-Mistake	0.295	0.001
SIE-Time	0.266	0.004
SIE-Mistake	0.325	<0.001
CTT A-Time	0.225	0.015
CTT B-Time	0.344	<0.001
CIE (B-A)	0.336	<0.001
CTT A-Mistake	0.047	0.616
CTT B-Mistake-Number	0.153	0.100
CTT B-Mistake-Color	0.117	0.209
BNT	−0.124	0.183
IADL	−0.199	0.031

Abbreviations: SIE-Time: Stroop interference effects—Time; SIE-Mistake: Stroop interference effects—Mistake; CIE (B-A): color trails test interference effects—(TimeB-TimeA); BNT: Boston naming test; IADL: instrumental activities of daily living scale.

**Table 5 brainsci-13-00616-t005:** Correlation between MIF level and WMH degree and cognitive function.

	r	*p*
Total Fazekas score	0.193	0.037
Total MoCA score	−0.316	0.001
Stroop D-Time	0.133	0.154
Stroop W-Time	0.151	0.104
Stroop C-Time	0.238	0.010
Stroop D-Mistake	−0.021	0.823
Stroop W-Mistake	0.110	0.239
Stroop C-Mistake	0.091	0.328
SIE-Time	0.186	0.045
SIE-Mistake	0.093	0.321
CTT A-Time	0.129	0.165
CTT B-Time	0.258	0.005
CIE (B-A)	0.304	0.001
CTT A-Mistake	0.158	0.089
CTT B-Mistake-N	0.133	0.154
CTT B-Mistake-C	0.146	0.117
BNT	−0.213	0.021
IADL	−0.126	0.176

Abbreviations: SIE-Time: Stroop interference effects—Time; SIE-Mistake: Stroop interference effects—Mistake; CIE (B-A): color trails test interference effects—(TimeB-TimeA); BNT: Boston naming test; IADL: instrumental activities of daily living scale.

**Table 6 brainsci-13-00616-t006:** ROC of serum MIF level and WMH degree in predicting CSVD-CI.

	AUC	Sensitivity	Specificity	95% CI	*p*
MIF (pg/mL)	0.661	0.723	0.596	0.561~0.761	0.003
Total Fazekas score	0.658	0.462	0.846	0.559~0.756	0.003

## Data Availability

Data are unavailable due to privacy or ethical restrictions.

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
