# Peer review of "The Relevance of Serum Macrophage Migration Inhibitory Factor Level and Executive Function in Patients with White Matter Hyperintensity in Cerebral Small Vessel Disease"

_brainsci, 2023, doi:10.3390/brainsci13040616_

Round 1
Reviewer 1 Report
The authors investigated potential associations between plasma MIF level and EF function, and between WMH severity (Fazekas rating) and EF function in those with clinically confirmed cSVD. EF was primarily assessed via Stroop and Color trails test. In addition, differences in WMH severity and MIF level were compared between participants divided into cognitively normal and cognitively impaired (determined by MoCA) groups. While potentially interesting, I have the following comments below.
Major Comments:
1) The motivation and novelty of the study is not clear to me in general. A better developed introduction could go a long way here. WMH severity may be associated with EF, but this is not really new, which the authors mention. While there may be reasons to look at the association of MIF with EF in cSVD, the reason for making this the primary focus of the current manuscript should be more thoroughly explained (for example, why not another plasma marker of vascular dysfunction or neurodegeneration). In the discussion, saying what the findings of this paper add to the overall literature would be helpful.
2) It is worth mentioning that I did not understand that the study was focused on studying MIF level and WMH as risk factors of EF by reading the abstract, it did not become clear until the final sentence of the introduction. The abstract could be adjusted to make this much clearer.
3) More details are definitely required in the Methods, particularly 2.2.3, but in general the study cannot be replicated from the information provided. Generally, I would encourage authors to avoid the excessive use of colons and semicolons in this section and go for a more traditional sentence and paragraph structure. In addition, how many of these measures were acquired is not really explained. Serum analysis needs more explanation.
a. Regarding 2.2.3, there are many important details which are simply not present. The reader is not told whether a 3T or 1.5 T MRI is used. Was a headcoil used and if so how many channels? What were the scan parameters for each scan? How were these 4 scans incorporated to determine Fazekas score? How was it determined if a WMH was in a PV or deep region? How many raters were used to determine WMH severity? Were they blinded to group? Was there high inter-rater reliability (if multiple raters)?
4) As far as I can tell the results are mostly fine if a bit lengthy. Authors might consider a correction for multiple comparisons given the tested associations with many different cognitive tests. It might also be helpful to the reader to either discuss all the group-related analysis first and then correlational analysis or vice versa.
5) I would objectively assess the English usage in the paper as moderate-to-good quality; but it could be improved to make the manuscript clearer overall.
Minor Comments
· Where it lists WMH metrics in Table 2, the p-value indicates there is a significant difference between groups, but the listed values are exactly the same in these groups. Is this an error?
· Since all participants had cSVD, did they also all have WMHs?
· The title for section 2.1 is “Objects”. Was this intended?
· It is not clear to me what “perfect” MRI imaging and clinical data means (line 73).
· On line 82, I think the authors meant “due to” rather than “with”.
· On line 88, since only the MoCA is used to determine group, I would divide into normal cognitive and impaired cognitive groups without referring to WMH, since MoCA cannot assess WMHs.
· Why were participants grouped by MoCA score and not EF function, since EF function is the focus of the study?
· On line 101, it would be useful to know exactly what is entailed in the “unified guidance in the same conditions” by the trained physician.
· I would avoid casual language in the discussion (lines 259 and 266).
· It might be more impactful to end the paper with the conclusion rather than the limitations.
Reviewer 2 Report
Excellent paper whose purpose was to to investigate the relationship between serum macrophage migration inhibitory factor (MIF) level and white matter hyperintensity (WMH) and executive dysfunction in patients with cerebral small vascular disease.
The methods used in the study, as well as its objectives, are clearly described. The statistical analysis is impressive.
I could conclude that it is an impressive study and I don't see any issues not to be published in current form.
Author Response
Thank you so much for your recognition and support of our team.
Reviewer 3 Report
Overall
This is an interesting study, but the writing style, including spaces, layouts, and formats, should be improved because the manuscript is difficult to read due to too many abbreviations, particularly in the Results (pages 4–7), which appear to be the study's highlights.
Please consider my major concerns, which are the comments in Nos. 1, 6, 7, 8, 11, and 12.
Abstract
1. The abstract should be rewritten to be more compact and concise, particularly in the Methods part, and a total of about 200 words should be the maximum.
2. Some of the abbreviations in the Results part should be replaced with full terms because they are first-time uses.
Introduction
3. This section is too brief and lacks reasoning to establish the significance of the study. Stronger evidence and reasonable justifications are necessary to add more.
Materials and Methods
4. In 2.1, please consider changing the term "objects" to "participants" in line 69.
5. In 2.2, this part needs more references, particularly in 2.2.2 and 2.2.3.
6. In 2.2.5, please indicate which tables or variables use which statistic methods for a better understanding since this part is too broadly written.
Results
7. For tables 1 and 2, please provide effect sizes, since some significant results showed no differences in comparisons between the two groups.
8. Please report actual significant P values to specify which results you are referring to in all of the corresponding texts in this section.
9. In table 3, all numbers should be at the same decimal point for consistency.
10. Please consider changing 0.000 to < 0.001.
11. In Figure 1, please give each picture a name (more information) and provide the corresponding text to describe what is important about these pictures for the study.
12. Overall, the results contained some interesting findings, but the manuscript did not clearly point them out or highlight them for the readers. The manuscript presented just plain and simple textual information.
Discussion
13. The first paragraph should move and be combined with the Introduction section on page 2.
14. There is some interesting interpretation within this section, such as in the third paragraph. However, more references cited within this section to support the study are recommended.
15. In lines 266–293, these should be reorganized. Some texts can be grouped as limitations and recommendations, and some should be separated into another section for the Conclusion.
16. Please provide information for the “Data Availability Statement” at the end of the manuscript.
Others
17. Please recheck and correct the information in lines 7–16 and 294–297 for "Author Contributions."
18. The use of "etc." such as in lines 80 and 102 should be omitted, and it should be described with full details.
Round 2
Reviewer 1 Report
I thank the authors for their thoughtful comments and revisions, and now recommend the manuscript for publication.
I have two further recommendations that the authors might consider, which I think would improve the manuscript, but are optional.
1) The T2-weighted scan parameters are now reported, but not the parameters of the others scans. It would be helpful to include the parameters for the other scans, if only as supplementary information.
2) This suggestion relates to a previous comment I had that is about the clarity of display in Table 2 . For example, from the reader's perspective, when looking at the row "Total Fazekas score", it looks like the NC group and IC group had exactly the same values (3.00; because they have the same median value), yet the p-value indicates they are significantly different. To increase clarify, that they are in fact different, I might report the median value to the nearest tenth value (2.9, 2.8 etc.), rather than the nearest integer value, so that the differences between groups can be better appreciated.
Reviewer 3 Report
The manuscript has improved significantly from the last version. Further minor modifications may be necessary to further enhance the manuscript.
1. The abbreviation “MIF” in the title might need to be replaced with its full term.
2. Rather than the use of “etc.” in lines 29, 84, 172, and 285, please consider removing or replacing them with details.
3. In lines 56–57, the study mentioned that “Since the pathogenesis remains unclear and the diagnosis of CSVD-CI remains controversial, ...” Please provide references for this statement.
4. The conclusion part (page 10) is too short. It can be improved to be longer with more study summaries and highlights.
